# The Effect of Incubator Cover on Newborn Vital Signs: The Design of Repeated Measurements in Two Separate Groups with No Control Group

**DOI:** 10.3390/children10071224

**Published:** 2023-07-14

**Authors:** Kenan Çetin, Behice Ekici

**Affiliations:** 1Neonatal Intensive Care Unit, Siverek State Hospital, Şanlıurfa 63600, Türkiye; kenanvecetin@gmail.com; 2School of Nursing, Maltepe University, Istanbul 34857, Türkiye

**Keywords:** term, preterm, incubator cover, light-dark cycle, vital signs

## Abstract

(1) Background: During their stays in neonatal intensive care units (NICU), newborns are exposed to many stimuli that disrupt their physiological indicators. The aim of this study was to investigate the impact of the light–dark cycle created with and without an incubator cover on the vital signs of term and preterm newborns. (2) Methods: A repeated measures design was used in the study utilizing two separate groups, without a control group. The study included 91 neonates hospitalized in a NICU (44 term and 47 preterm). With and without an incubator cover, the newborns’ vital signs (heart rate (HR), respiratory rate (RR), oxygen saturation (SpO_2_), and body temperature (BT)) were measured. Three separate measurements were taken. (3) Results: The mean age of the newborns was 37.0 weeks. There was no significant difference between the HR and RR medians of the term and preterms in the incubator undraped and clad measurements (*p* > 0.05). At the first measurement, the SpO_2_ medians of the incubator-covered term and preterms were significantly higher than those of the incubator-covered term and preterms (*p* = 0.001). (4) Conclusions: The vital signs of the neonates demonstrated variable responses in the measurements when their incubators were covered vs. when they were not covered. However, more research on the effect of the light-dark cycle on their vital signs is required.

## 1. Introduction

The ability of term and preterm newborns to respond to external stimuli is insufficiently developed [1]. Term and preterm newborns are subjected to a variety of stimuli that disrupt their physiological indicators, particularly while in a neonatal intensive care unit (NICU). The working and alarm sounds of devices such as mechanical ventilators and monitors, the voices of employees in the unit, the sounds of opening and closing incubator covers, continuous ambient lighting, and intense light are some of these stimuli [2,3,4,5,6,7]. These stimuli are said to cause deviations in newborns’ physiological indicators, hospitalization lengthening, and delays in their growth and development, as well as hearing and vision loss [6,8,9,10,11,12]. In one study, it was discovered that extremely premature newborns exposed to 332 light levels perceived extremely low light levels [13]. It is recommended that cyclical lighting be used in NICUs, with a light level of 10–600 lux [6,12]. According to studies, intense and continuous lighting causes stress responses, sleep pattern deterioration, tachycardia, tachypnea, increases in oxygen consumption and motor activity, a decrease in oxygen saturation (SpO_2_), neurophysiological changes, and a prolongation of discharge time [6,7,11,14,15]. Developmental care practices should be included in newborn care plans to prevent these problems, support newborn neurodevelopment, and reduce the stimuli in NICUs [1,2,6,15,16,17,18,19].

Baby sleep eye patches, earplugs, helmets, and incubator covers designed in various shapes and designs are used in the developmental care practices of NICUs to reduce the signals that disturb the circadian rhythm, such as sound and light [1,15,18,20,21,22,23,24]. Studies have shown that incubator and face covers reduce the light exposure of newborns [22,24], reduce the pain scores of preterms [25], reduce noise level and stress symptoms [26], improve the sleep quality of clinically stable preterms for a short period of time [21], and increase the sleep duration of newborns [27].

In the literature review, however, there were no studies examining the effect of incubator covers on the vital signs of full-term and/or premature newborns. In Turkey, there is no protocol for using incubator covers in NICUs. Similarly, in the NICU where the research was conducted, no incubator covers were used, and there were no incubator covers in the unit. In this study, a cover designed by the researchers was used to minimize the newborns’ exposure to light while in the NICU incubator (Figure 1). The authors of the study investigated the effect of this incubator cover on the vital signs of term and preterm neonates who met the criteria for the sample selection (Figure 2).

In this study, the following hypothesis was tested:

**H1.** 
*An incubator cover contributes to maintaining term and/or preterm newborns’ vital signs within the reference ranges.*


## 2. Materials and Methods

### 2.1. Study Design and Population

#### 2.1.1. Study Design

In this study, a design consisting of repeated measurements in two distinct groups (1st measurement–2nd measurement–3rd measurement) was used to investigate the changes within the same sample group. ClinicalTrials.gov (accessed on 28 May 2023) Identifier: NCT05892809.

#### 2.1.2. Population

The population of the study consisted of 195 newborns hospitalized in a second-level NICU of a public hospital in a district of southeastern Turkey. The study sample included 91 newborns who met the inclusion criteria. According to their gestational age, the infants were categorized as either full-term (*n* = 44) or preterm (*n* = 47). The study was conducted separately on the term group and preterm group. A post hoc power analysis on the 91 newborns yielded a value of 0.998, with an alpha error of 0.05 (G*Power 3.1.9.2).

#### 2.1.3. Inclusion and Exclusion Criteria

The inclusion criteria were: (1) being in the first week of life, (2) having a gestational age of 24–42 weeks, (3) a weight of ≥1500 g, (4) being treated in an incubator, and (5) the parents providing verbal and written consent for being included in the study. The exclusion criteria for the study included: (1) receiving invasive mechanical ventilation support, (2) organ failure, and (3) the need for surgery. Newborns who died shortly after admission to the NICU (*n* = 9), those who were intubated (*n* = 23), those with a gestational age of >42 weeks (*n* = 17), and those whose parents could not be reached to obtain consent for their participation in the study (*n* = 55) were excluded. Figure 3 displays the CONSORT flowchart for the study.

### 2.2. Data Collection

#### 2.2.1. Procedure

The intervention and inclusion of newborns in the study began on the first day of the participants’ admission to the NICU. The demographic information and medical history of the newborns were obtained from their patient files. The newborns’ vital signs were recorded during periods when they were calm and awake in the supine position. Their vital signs were measured in the following order: heart rate (HR), SpO_2_, body temperature (BT), and respiratory rate (RR). These vital signs were firstly measured three times without the incubator cover (at 0, 15th, and 30th min), and then three times at the same time intervals with the incubator cover in place. The newborn information and follow-up form (NIFF) were used to record all the data. The second researcher measured all the vital signs and recorded all the necessary data on the NIFF.

#### 2.2.2. Neonate Information and Follow-Up Form (NIFF)

The authors of the study created the NIFF. The descriptive characteristics of the newborns (gestational age, gender, body weight, and number of days in the NICU) were recorded in the first part of the form, and the vital signs of the newborns (HR, RR, SpO_2_, and BT) were recorded in the second part. Each vital sign was written on four lines. Across the rows, three columns were created, in which the newborns’ first, second, and third measurement results were written, with and without the incubator cover.

#### 2.2.3. Measurement of Vital Signs

The HR and SpO_2_ were measured using a console-type Nellcor Bedside pulse oximeter. The pulse oximetry probe’s sensor apparatus was placed on the newborn’s right wrist for the HR and SpO_2_ measurements [28,29,30]. After waiting for the results to appear on the device’s screen, the HR and SpO_2_ values displayed on the screen were recorded. For the BT measurement, a multifunctional Kangaroo KR 1000 model device was used. The skin temperature probe was placed on the newborn’s left wrist [31,32]. The device was placed in the skin control mode for the BT measurement, and the BT value displayed on the screen was recorded. The researcher used a clock stopwatch to time the RR measurement. The investigator counted the newborn’s breathing (the rise and fall of the chest and abdomen was considered one) for one minute (min) and recorded this number as the RR value. The front side of the incubator drape was opened for only one minute during the RR measurement with the incubator drape (Figure 1). The incubator cover was removed for all the other measurements. All the vital sign measurement results were recorded on the NIFF.

#### 2.2.4. Incubator Cover

Since an incubator cover was not used or present in the NICU where the research was conducted, this incubator cover was designed by the researchers and sewn in the sewing department of the hospital that was utilized in this study (Figure 1). The quilting technique was used to sew the incubator cover, which was designed in three layers to provide light insulation. The upper layer of the incubator cover was made of dark green 100% cotton fabric, the lower layer was made of white fabric, and the middle layer was filled with cotton. In order to accommodate the dimensions of the incubators used in the NICU, the incubator cover was designed as a 5-piece set (top piece, two side pieces, one front piece, and one back piece). A zip was sewn between the front, back, and two side pieces to connect them. These parts were designed to open at the top of the incubator and close at the bottom. This design allowed for only the required part of the incubator cover to be opened when necessary.

### 2.3. Statistical Analysis

For the data analysis, SPSS Windows version 15.0 (SPSS Inc., Chicago, IL, USA) was utilized. The Kolmogorov–Smirnov test was used to examine the data distribution. Number, percentage, mean, standard deviation, median, and interquartile range (IQR) were used to assess the descriptive characteristics of the newborns. The vital sign measurement results were evaluated using the median and IQR (25–75%. values). The descriptive properties of the term and preterms were compared using the Pearson Chi-square (χ^2^) and Mann–Whitney U (*M–U*) tests. The Wilcoxon signed-row test was used to investigate the effect of the incubator cover on the newborns’ vital signs. The Friedman χ^2^ test was used to examine the changes in the repeated vital sign measurements. By calculating the Cohen effect value, the magnitude of the effect of the difference in the results of the analyses was determined. The Cohen effect size was interpreted using intervals (≤0.2 low effect; 0.21–0.50 low effect; 0.51–1.00 moderate effect; and >1.00 strong effect) [33]. For statistical significance, *p* < 0.05 was accepted.

### 2.4. Ethical Considerations

The study was approved by the Ethics Committee of Maltepe University (EKK/2017/83). An informed consent form was obtained from the parents of the infants who met the inclusion criteria. In this form, the purpose of the research, the method by which it would be conducted, the incubator covers, and their potential effects were written in clear and simple language. Furthermore, the parents were informed verbally and in writing that they could opt out of the study at any time.

## 3. Results

In the study, the data from a total of 91 newborns were analyzed. In total, 48.4% of the newborns were full-term, while 51.6% were premature. The median gestational age of all the newborns was 37.0 weeks, the median body weight was 3.280 g, and the median length of the NICU hospitalization was 4.0 days. The causes for the admission of the neonates to the NICU were hypoglycemia (term, *n* = 22; preterm, *n* = 27), dehydration (term, *n* = 8; preterm, *n* = 12), and electrolyte deficiency (term, *n* = 14; preterm, *n* = 8) (Table 1). All the newborns were admitted to the NICU within their first week of life and immediately after birth.

When the term neonates in covered (medians (IQRs): 1st = 133 (127–142), 2nd = 131.2 (100–160), or 3rd = 133.9 (96–157) measurements) or uncovered (medians (IQRs): 1st = 135.5 (124.5–136), 2nd = 131.1 (107–147), or 3rd = 133.3 (111–161) measurements) incubators were compared, the median HRs did not differ in the 1st (*p* = 0.208), 2nd (*p* = 0.410), or 3rd (*p* = 0.810) measurements (*p* > 0.05). When the serial measurements were compared between themselves, the HRs were similar for the uncovered (*p* = 0.127) and covered (*p* = 0.227) incubators in the term neonates (*p* > 0.05). Furthermore, when the preterm neonates in the covered (medians (IQRs): 1st = 135 (127–144), 2nd = 132 (127–141), or 3rd = 135 (127–145) measurements) or uncovered (medians (IQRs): 1st = 136 (126–144), 2nd = 136 (128–144), or 3rd = 136 (127–143) measurements) incubators were compared, their median HRs did not differ in the 1st (*p* = 0.946), 2nd (*p* = 0.194), or 3rd (*p* = 0.838) measurements (*p* > 0.05). When the serial measurements were compared between themselves, the HRs were similar for the uncovered (*p* = 0.639) and covered (*p* = 0.517) incubators in the preterm neonates (*p* > 0.05) (Figure 4).

When the term neonates in the covered (medians (IQRs): 1st = 50 (47–52.5), 2nd = 50.5 (48–54), or 3rd = 50 (48–54) measurements) or uncovered (medians (IQRs): 1st = 50 (48–54), 2nd = 51 (48–52), or 3rd = 50 (48–53) measurements) incubators were compared, their median RRs did not differ in the 1st (*p* = 0.466), 2nd (*p* = 0.987), or 3rd (*p* = 0.635) measurements (*p* > 0.05). When the serial measurements were compared between themselves, the RRs were similar for the uncovered (*p* = 0.338) and covered (*p* = 0.950) incubators in the term neonates (*p* > 0.05). Furthermore, when the preterm neonates in the covered (medians (IQRs): 1st = 55 (52–59), 2nd = 56 (52–60), or 3rd = 54 (50–60) measurements) or uncovered (medians (IQRs): 1st = 54 (54–54), 2nd = 56 (52–56), or 3rd = 55 (52–58) measurements) incubators were compared, their median RRs did not differ in the 1st (*p* = 0.171), 2nd (*p* = 0.344), or 3rd (*p* = 0.678) measurements (*p* > 0.05). When the serial measurements were compared between themselves, the RRs were similar for the uncovered (*p* = 0.932) and covered (*p* = 0.164) incubators in the preterm neonates (*p* > 0.05) (Figure 5).

When the term neonates in the covered (medians (IQRs): 1st = 99 (96.5–100), 2nd = 99.5 (98–100), or 3rd = 99 (97.5–100) measurements) or uncovered (medians (IQRs): 1st = 100 (100–100), 2nd = 99 (96.5–100), or 3rd = 99 (98–100) measurements) incubators were compared, the median SpO_2_ did not differ in the 2nd (*p* = 0.596) or 3rd (*p* = 0.542) measurements (*p* > 0.05), but was significantly different in the 1st measurement (*p* = 0.001). When the serial measurements were compared between themselves, the SpO_2_ was similar for the uncovered (*p* = 0.227), yet significantly different for the term neonates in the covered (*p* = 0.001) incubators. Furthermore, when the preterm neonates in the covered (medians (IQRs): 1st = 99 (98–100), 2nd = 100 (98–100), or 3rd = 99 (97–100) measurements) or uncovered (medians (IQRs): 1st = 100 (100–100), 2nd = 99 (99–100), or 3rd = 99 (98–100) measurements) incubators were compared, their median SpO_2_ was also significantly different in the 1st (*p* = 0.001) measurement, yet similar during the 2nd (*p* = 0.441) or 3rd (*p* = 0.576) measurements (*p* > 0.05). When the serial measurements were compared between themselves, the SpO_2_ was similar for the uncovered (*p* = 0.679), yet significantly different in the preterms in the covered incubators (*p* = 0.001) (Figure 6).

When the term neonates in the covered (medians (IQRs): 1st = 36.5 (36.3–36.7), 2nd = 36.5 (36.4–36.7), or 3rd = 36.5 (36.4–36.7) measurements) or uncovered (medians (IQRs): 1st = 36.4 (36.4–36.4), 2nd = 36.3 (36.3–36.5) or 3rd = 36.5 (36.3–36.7) measurements) incubators were compared, their median BTs did not differ in the 1st (*p* = 0.137) or 3rd (*p* = 0.288) measurements (*p* > 0.05), but were significantly different in the 2nd measurement (*p* =.006). When the serial measurements were compared between themselves, the BTs were similar in the uncovered (*p* = 0.893) and covered (*p* = 0.144) term neonates (*p* > 0.05) Furthermore, when the preterm neonates in the covered (medians (IQRs): 1st = 36.5 (36.4–36.8), 2nd = 36.4 (36.3–36.7), or 3rd = 36.5 (36.3–36.7) measurements) or uncovered (medians (IQRs): 1st = 36.4 (36.4–36.5), 2nd = 36.4 (36.3–36.7), or 3rd = 36.5 (36.4–36.7) measurements) incubators were compared, their median BTs were also significantly different in the 1st (*p* = 0.021) measurement, yet similar during the 2nd (*p* = 0.887) or 3rd (*p* = 0.906) measurements (*p* > 0.05). When the serial measurements were compared between themselves, the BTs were similar for the uncovered (*p* = 0.408), yet significantly different in the preterms in the covered incubators (*p* = 0.034) (Figure 7).

## 4. Discussion

During their time in a NICU, term and preterm infants are exposed to various stimuli. There are a variety of care practices and materials used to reduce these stimuli and promote the establishment of circadian rhythm and the neurodevelopment of newborns. An incubator cover is one of these materials [6,14,16,17,22,23,24]. A literature search reveals only a few studies on the effect of these incubator covers on newborns. For example, incubator covers designed by researchers with light and sound insulation properties have been found to reduce the noise levels in the incubator and decrease the stress symptoms of preterms [26], decrease the pain scores of preterms [25], improve the sleep quality of preterms for a short time [21], and increase sleep duration [27]. We were unable to find any study examining the effect of an incubator cover on the vital signs of full-term and premature infants. In this study, vital signs, one of the most important indicators of newborn clinical status, were measured three times with and three times without an incubator cover. The measurement results were discussed alongside vital sign reference ranges and other research findings.

HR provides information on a newborn’s cardiovascular function. Immediately after birth, a newborn’s HR can reach up to 180 beats per minute. In the days that follow, the HR ranges between 100–180 beats per minute while awake and 80–160 beats per minute while sleeping, and this value rises significantly in situations such as crying and stress [34,35,36]. In one study, the median heart rate of term and preterm newborns was reported to be 157 beats per minute 10 min after birth [30]. In two studies examining the HR in healthy term newborns in the hours immediately after birth, the median HRs were measured as 127 beats/min [37] and 121 beats/min [38]. It has been reported that the HR in the postnatal days of non-hospitalized newborns is in the range of 100–125 beats/min [39]. The HR of newborns monitored in a NICU increases or decreases depending on health problems, neurodevelopmental levels, and the intensity and continuity of light in the NICU [6,11,12]. The median HR (131.2–136 beats/min) of all the newborns in this study was higher than the normal mean HR (100–125 beats/min) of healthy newborns expected in the first postnatal days. When the incubator was covered, the line graph of the term neonates showed a decrease in the second measurement and an increase of 2 units in the third measurement (1st measurement > 3rd measurement > 2nd measurement). When the incubator was not covered, the second measurement decreased and then increased (3rd measurement > 1st measurement > 2nd measurement). However, there was no statistically significant difference between these measurements (*p* > 0.05). The measurement result in the line graph of the median HR of the preterms did not change when the incubator was covered (1st measurement = 2nd measurement = 3rd measurement). When the incubator was not covered, it showed a sharp decrease (3 units) in the second measurement and then increased to the first measurement value in the third measurement (1st measurement = 3rd measurement > 2nd measurement). However, there was no statistically significant difference between these measurements either (*p* < 0.05) (Figure 4). According to these findings, the newborns responded to the dark environment provided by the incubator cover, and the HR medians, albeit at a low level, changed in both directions. According to the literature, the light level in a NICU should be in the range of 10–600 lux [6]. However, one study reported that very preterm newborns exposed to 332 light levels had an increased HR of 3.8 beats per minute, despite the fact that the light levels were within the recommended range [13]. Because newborns, particularly immature newborns, respond to even very small changes in light levels, appropriate methods and materials should be utilized as much as possible to reduce the light level in a NICU to the lower limit of the level recommended in the literature. In a study on preterms with an average gestational age of 32 weeks, evaluating the effect on weight gain of a dark environment provided by the use of helmets, the mean daily HR of the preterms that stayed in a dark environment was reported to be 149.9 ± 0.73 compared to 150.9 ± 0.46 for those who stayed in an environment with continuous light [20]. The mean HR of the term and preterm infants included in this study was lower than that of Sánchez-Sánchez et al.’s study, with a value close to the upper limit of the normal HR range (100–125 beats/min). The lower mean HR in this study, when compared to Sánchez-Sánchez et al.’s study, was attributed to the use of incubator covers and newborns with a higher gestational age and birth weight. In a study examining the effect of an incubator cover with sound insulation properties on preterms with a mean gestational age of 32 weeks, the authors reported that the mean HR in the first minute (135.5 ± 13.37) was approximately four units lower (131.63 ± 11.95) in the second measurement taken 30 min later [26]. Although the gestational age and body weight of the newborns in this study were greater than those of the preterm infants in Karadağ and Balcı’s study, the reduction in HR was attributed to the use of incubator covers with light and sound insulation properties.

The median RR of the term newborns was reported as 44 breaths per minute [37] immediately after birth and 37 breaths per minute [38] in the subsequent hours. In one study, the RR of healthy neonates was found to range between 30 and 40 breaths per minute [39]. After the physiological systems of a newborn have stabilized, the RR stabilizes between 30 and 60 breaths per minute [35,36], with a mean of 35 breaths per minute [36]. However, due to various health issues, NICU-hospitalized newborns continue to exhibit irregular breathing. In a study examining the effect of light level on extremely premature newborns in a NICU, it was found that the RR of the newborns decreased proportionally to a decrease in the light level [13]. The newborns who took part in this study had a median RR of 50–56 breaths per minute, which falls within the normal RR reference range of 30–60 breaths per minute. With and without the incubator being covered, the RR medians of the term newborns increased at the second measurement, and then decreased to the first measurement at the third measurement (2nd measurement > 1st measurement = 3rd measurement). On the line graph, the RR medians of the preterms rose at measurement 2 with and without the incubator covered, and they fell once more at measurement 3 (incubator covered: 2nd measurement > 3rd measurement > 1st measurement; incubator uncovered: 2nd measurement > 1st measurement > 3rd measurement). However, there was no significant difference between these measurements (*p* > 0.05) (Figure 5). In a study, preterms with sound-insulating incubator covers had RRs at 30 min (41.95 ± 9.86) that were significantly lower than those without covers (45.16 ± 10.48) [26]. In another study of premature newborns, where a dark environment was created using helmets, the RR of those who wore helmets (55.9 ± 0.22) vs. those who did not (55.7 ± 0.16) were found to be similar [20]. According to the findings of this study, the order of these materials, in terms of their positive effect on the RR of the preterms, was: incubator cover with sound insulation feature, incubator cover with light insulation feature, and helmets.

The SpO_2_ level is one of the physiological indicators of the clinical condition of newborns. In the first minute after birth, SpO_2_ levels are reported to be between 60 and 65%, and between 85 and 95% by the tenth minute. In addition, it is advised to maintain these SpO2 levels between 90 and 95% in newborns with a gestational age of less than 28 weeks [34,35,40,41]. In one study, the SpO_2_ level of newborns with a gestational age of 38 ± 4 weeks was over 90% at all the measurements between the 5th and 10th minutes [29]. In a study conducted with healthy newborns, the SpO_2_ level of newborns with stable physiological systems was reported to be above 95% [39]. In one study, it was stated that the SpO_2_ monitoring results of preterm newborns were within the recommended range (90–95%) [42]. Considering the newborns’ gestational age (median 37.0 weeks) and body weight (median 3.280 g), the SpO_2_ values found in our study were consistent with those in the literature (≥99%). The median SpO_2_ of the incubator-covered term newborns was significantly higher than that of the non-incubator-covered term newborns in the first measurement (*p* = 0.001). The difference in this measure, the incubator cover, had a moderate Cohen effect size (*r* = 0.65). Similarly, the median SpO_2_ of the incubator-covered preterms was significantly higher than the preterms without incubator covers (*p* = 0.001). The difference in this measure, the incubator cover, had a moderate Cohen effect size (*r* = 0.77). In the repeated measurements, there was a significant difference between the median SpO_2_ of the three measurements for the term and preterm infants when the incubator was covered (*p* = 0.001). This difference was due to the fact that the medians of the SpO_2_ in measurement 1 were higher than those in the other two measurements (1st measurement > 2nd measurement = 3rd measurement) (Figure 6). The SpO_2_ averages of the preterms with and without sound-insulating incubator covers were 96% in the first minute and 97% in the 30th minute in a study by Karadağ and Balcı [26]. These findings show that two incubator covers with light and sound insulation properties did not change the SpO_2_ levels of newborns within the reference ranges and helped them to maintain their physiological functions.

Body temperature is another physiological indicator of a newborn. Newborns’ BTs decrease by 0.1 °C immediately after birth, leading to hypothermia in some neonates [35]. In the days that follow, newborns’ body temperatures should be kept between 36.5 °C and 37.5 °C, with an average of 37 °C [34,36,43]. However, postnatal hypothermia problems may persist, because premature newborns’ thermoregulation abilities are not fully developed [35,44]. In a study of premature newborns in their first week of life, the mean abdominal temperature was reported as 36.5 °C, and foot temperature as 35.7 °C [45]. The BT medians in the left wrist measurement of the newborns in their first week of life were found to be between 36.3 and 36.5 °C in the term newborns and between 36.4 and 36.5 °C in the preterm newborns in this study. These BT values are consistent with the literature. At the same time, the abdominal temperatures measured in Knobel-Dail et al.’s study were close to or the same as the preterms’ left wrist BT median values in this study. In this study, the median BT of the incubator-covered terms was lower in the second measurement than the medians of the non-incubator-covered term newborns (*p* = 0.006). The Cohen effect size, on the other hand, had a minor effect in favor of not having an incubator cover (*r* = 0.41). The median BT of the incubator-covered preterms was lower than the median of the incubator-covered preterms in the first measurement (*p* = 0.02). The Cohen effect size of this difference, however, had a small effect in favor of not having an incubator cover (*r* = 0.33). There was a significant difference in the BT medians of the three measurements in the line plot of the preterms with the incubator covered (*p* = 0.03). This difference was due to the fact that the BT medians in measure 3 were higher than those in the other two measures (3rd measurement > 2nd measurement = 1st measurement) (Figure 7). According to Sánchez-Sánchez et al., the average daily BT of premature newborns who wore helmets was 37.02 ± 0.02, and 37.5 ± 0.01 for those who were constantly exposed to light [20]. According to these findings, the dark environment provided by an incubator cover and helmet caused a small increase and decrease in the BTs in the newborns, with some measurements remaining unchanged.

The vital sign measurement results in this study confirmed hypothesis H_1_: (incubator covers contribute to maintaining the vital signs of term and/or preterm newborns within the reference ranges).

## 5. Strengths and Limitations

In this study, developmental supportive care employed a material that is simple to use and does not cause stress reactions because the infant is not touched. Pulse oximetry and temperature probes were used in the repeated measurements of the vital signs in the study to prevent the negative effects of potential stimuli during the measurements; the values were displayed on the monitor. However, the RR was determined by the investigator by counting breaths, because there was no monitor in the NICU with an RR display parameter. The vital signs may have been affected by the NICU staff’s talking or the alarm sounds of the devices. Due to the high number of newborns in the NICU at the hospital where the study was conducted, more infants were unable to be included in the study. The fact that the study only included clinically healthy newborns and that the monitoring and recording of their vital signs was conducted for a brief period of time is a limitation.

## 6. Implications for Practice

There are always fluctuations in the vital signs of newborns in a NICU. These alterations must be carefully monitored. Additionally, developmental supportive care practices should be emphasized to prevent unintended effects of these changes. It will be beneficial to discuss the developmental supportive care practices and materials that can be used for this purpose with neonatal intensive care nurses, as well as to emphasize the significance of evidence-based practices based on current research. One of the practices of developmental supportive care is reducing the amount of light in a NICU and increasing the amount of time newborns spend in the dark. Materials such as baby sleep eye patches and incubator covers can be used for this purpose. In addition, developmental care procedures should be implemented to reduce newborns’ exposure to all stimuli in a NICU.

## 7. Conclusions

In conclusion, the vital signs of neonates with a stable clinical status were altered by the light–dark cycle. However, additional data are required regarding the effect of the light–dark cycle on the increases or decreases in the newborn vital signs. For this reason, multicenter studies monitoring the vital signs of newborns hospitalized at various NICU levels are required. In these studies, alternate materials can be used in randomized control designs, and repeated measurement designs can be used to track changes in the same sample group.

## Figures and Tables

**Figure 1 children-10-01224-f001:**
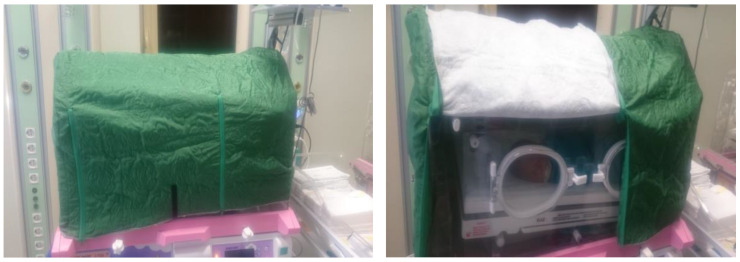
Cover for incubators designed by researchers.

**Figure 2 children-10-01224-f002:**
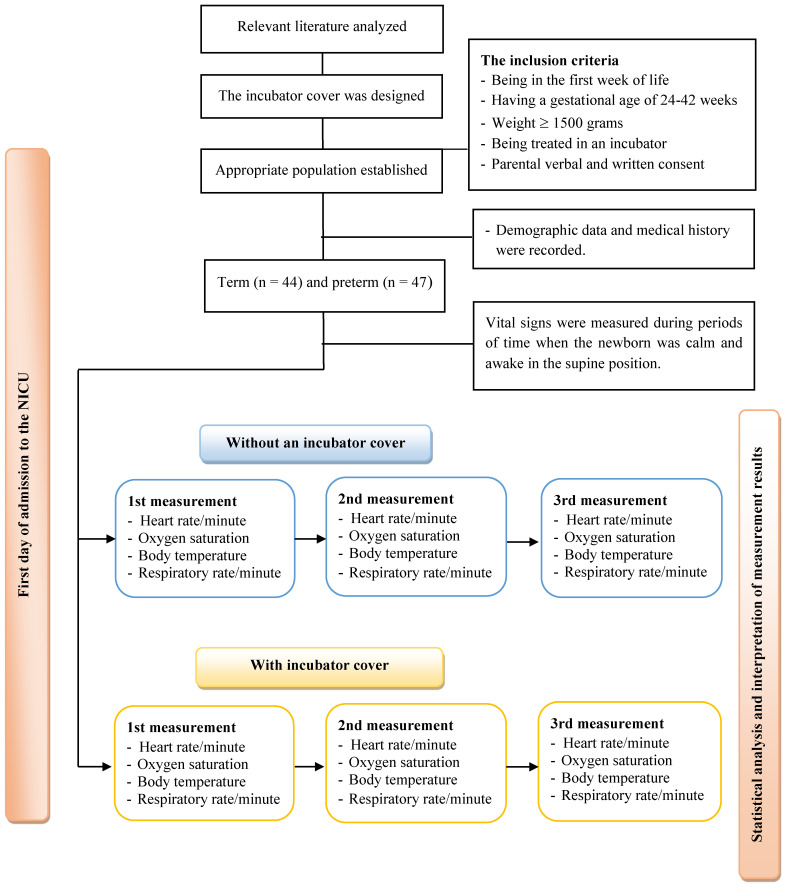
Graphical abstract of this study.

**Figure 3 children-10-01224-f003:**
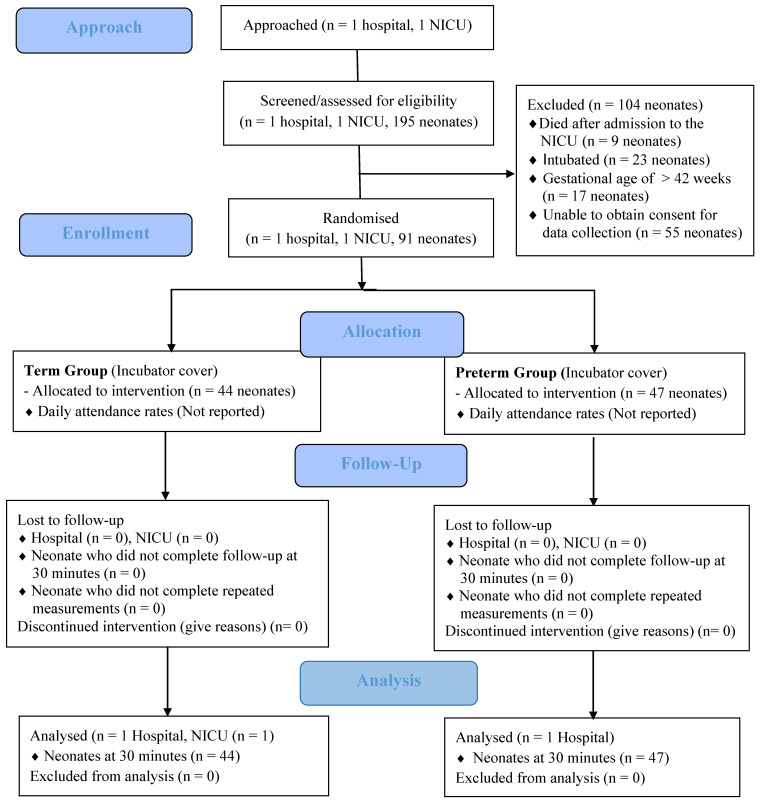
CONSORT diagram of this study.

**Figure 4 children-10-01224-f004:**
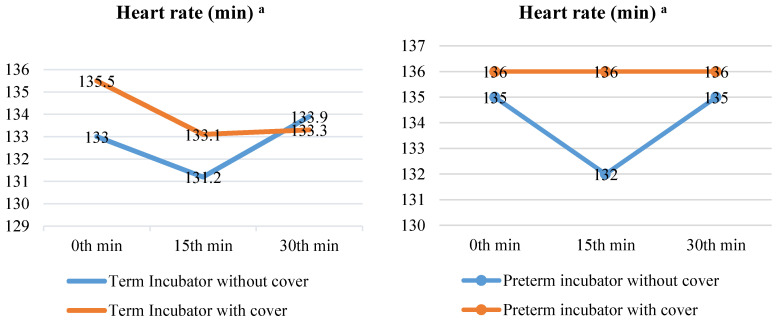
Comparison of median heart rate of newborns with and without incubator cover. ^a^ Heart rate at 1st min, Abbreviations: min = Minute. Notes: The median pulse rates at 0 min, 15 min, and 30 min are provided. In repeated measurements, the first measurement was taken at the 0th min, the second at the 15th min, and the third at the 30th min. The Friedman (χ^2^) test was used to examine the variation in repeated pulse rate measurements.

**Figure 5 children-10-01224-f005:**
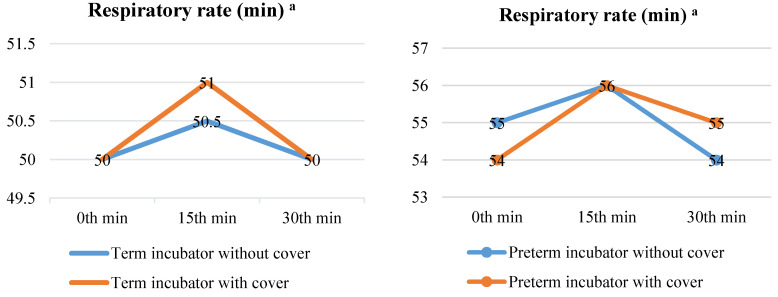
Comparison of median respiratory rate of newborns with and without incubator cover. ^a^ Respiratory rate at 1st min, Abbreviations: min = Minute. Notes: The median respiratory rates at 0 min, 15 min, and 30 min are provided. In repeated measurements, the first measurement was taken at the 0th min, the second at the 15th min, and the third at the 30th min. The Friedman (χ^2^) test was used to examine the variation in repeated respiratory rate measurements.

**Figure 6 children-10-01224-f006:**
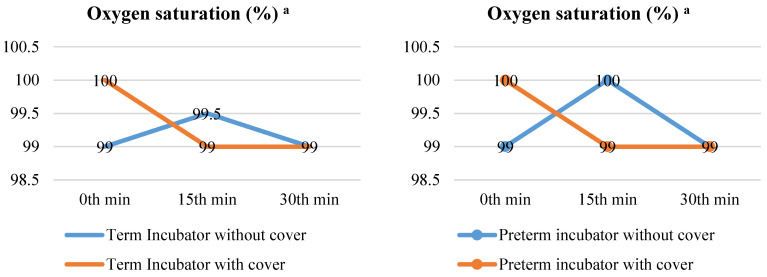
Comparison of medians of oxygen saturation of newborns with and without incubator cover. ^a^ Oxygen saturation at 1st min, Abbreviations: min = Minute. Notes: The median oxygen saturation at 0 min, 15 min, and 30 min are provided. In repeated measurements, the first measurement was taken at the 0th min, the second at the 15th min, and the third at the 30th min. The Friedman (χ^2^) test was used to examine the variation in repeated oxygen saturation measurements.

**Figure 7 children-10-01224-f007:**
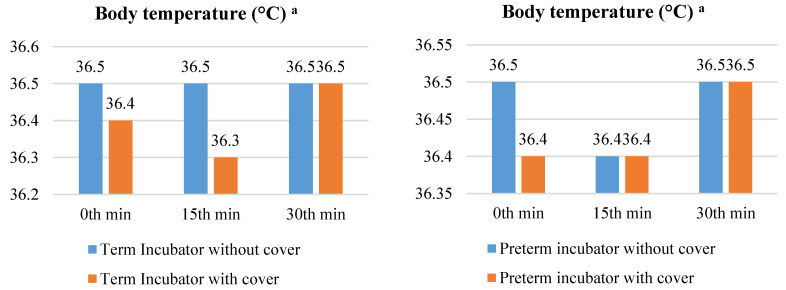
Comparison of median body temperature of newborns with and without incubator cover. ^a^ Body temperature at 1st min, Abbreviations: min = Minute. Notes: The median body temperature at 0 min, 15 min, and 30 min are provided. In repeated measurements, the first measurement was taken at the 0th min, the second at the 15th min, and the third at the 30th min. The Friedman (χ^2^) test was used to examine the variation in repeated body temperature measurements.

**Table 1 children-10-01224-t001:** Characteristics of newborns (*n* = 91).

Characteristics	Term (*n* = 44) (37–41 wk)	Preterm (*n* = 47) (32–36 wk)	Test; *p*-Value
Gender, *n* (%)			*^a^ 0.132; 0.716*
Female	18 (40.9)	21 (44.7)	
Male	26 (59.1)	26 (55.3)	
Gestational age, *n* (%)	44 (48.4)	47 (51.6)	
Weight, g, median (range)	3.440 (3.270–3.860)	3.000 (2.500–3.300)	*^b^ −4.186; 0.001*
The causes for admission to NICU			
Hypoglycemia	22 (50)	27 (57.4)	
Dehydration	8 (18.2)	12 (25.5)	
Electrolyte deficiency	14 (31.8)	8 (17.1)	
Length of NICU hospitalization, day, median (IQR)	3.0 (3.0–4.0)	4.0 (2.0–5.0)	*^b^ −0.648; 0.517*
^c^ Length of NICU hospitalization, day, mean ± SD = 3.7 ± 1.6; median = 4.0			
^c^ Gestational age, week, mean ± SD = 37.2 ± 1.4; median = 37.0			
^c^ Weight, g, mean ± SD = 3152.8 ± 881.5; median = 3.280			

Abbreviations: *n* = Number, % = Frequency, Gestational age = Week, wk = Week, g = Gram, SD = Standard Deviation, and IQR = Interquartile range (25th and 75th between). Bold data indicate *p* < 0.05. ^a^ Pearson χ^2^ test. ^b^ Mann–Whitney *U* test. ^c^ For all neonates.

## Data Availability

Data sharing is not applicable to this article.

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
