# Peer review of "The Effect of Incubator Cover on Newborn Vital Signs: The Design of Repeated Measurements in Two Separate Groups with No Control Group"

_children, 2023, doi:10.3390/children10071224_

Round 1
Reviewer 1 Report
Term babies why were they nursed in incubator and not radiant warmer, please explain. Inclusion criteria shows >24 weeks and weight>1500gms but table 1 shows recruited babies>32 weeks, weight from 2.5kgs. Please explain. What is the viability gestation for preterm babies in your NICU. Median HR, RR, Spo2 was all normal values, please mention the IQR. Please mention the cause for admission to NICU of the included babies. Duration of NICU stay was very less for these babies, explain.
English language minor corrections are required. Female spelling in table 1 is grossly wrong.
Reviewer 2 Report
The conclusion in the abstract is confusing. It should speak specifically to what increased or decreased that is relevant.
I have some doubts about the statistical significance calculations used in determining in a small sample size that a difference of 1% in oxygen saturations was significant at time point 1 but the same difference at time point 2 for preterm infants is not anywhere near significant.
I am also concerned that there is little reflection on the clinical significance of any of these findings. The differences even if statistically significant are not clinically significant and there should be ample reflection on this.
English is in need of some minor grammatical changes
Reviewer 3 Report
The article is a is a prospective observational study with preterm and term infants looking at changes in vital signs as a function of time against the presence or absence of incubator covers. While interesting, the article has parts that are redundant especially in the discussion section and some of the practices used in the study needs further justification/explanation.
I have the following concerns and comments:
1) A major concern includes the use of covered incubators for ‘term’ infants. Can the author justify why term infants were kept inside the incubator? Generally, incubators are reserved for only preterm infants as their skin is immature and they have a harder time controlling their temperature with high risk for hypothermia and dehydration from fluid loss.
2) While the inclusion criteria included preterm infants > 24 wks and >1500gms, preterm infants included had an average weight of 3000gms and GA range from 32-36wks. This complicates the interpretation of the study. Can the author reasonably provide argument how this preterm infant population qualifies as a overall representative sample for infants with GA ranging from 24-36 weeks?
3) Line 66: what is the reference range that the author is alluding to? Needs to provide reference and include the reference range that was being used.
4) There are redundancies in the discussion portion (e.g., Line 268 – 271, line 341 – 346) as the results are repeated. Discussion section should only critique and discuss the interpretation of the results from the study. Thus, a large portion of the discussion section should be re-written.
5) Line 358: It is unclear what the author means by BT drops to 33-34*C in all newborns. This is not true and would be considered hypothermia. This sentence needs to be paraphrased.
6) Line 18: ‘Increase or decrease’ – suggest changing this to ‘variable response in neonatal vital signs’.
I did not have any issue following the English language in the article
Reviewer 4 Report
This paper has great clinical importance, however, it is not clearly presented. The general quality of the manuscript should be improved. The authors need to edit it carefully. Figures and tables have marginal quality and need substantial further editing.
In the introduction or discussion sections, authors need to discuss circadian rhythm and its association with birth and infant outcomes (PMID: 36266474; PMID: 27984231, etc )
Minor: The name of the first author is written twice while the name of the second author is missing.
Round 2
Reviewer 1 Report
Light dark cycle has been studied on the preterm weight gain and development of preterm neonates. Limitations of the study is that only stable babies were included in the study and vitals recorded for a short duration of time.Kindly modify the conclusion.
Author Response
Thank you very much again for your comments and contributions.

Reviewer 2 Report
Thank you for addressing the my comments
Author Response

(The authors gave the same response as above.)

Reviewer 3 Report
I am satisfied with the author's responses. Not other comments. Congratulations to the authors!
Author Response

(The authors gave the same response as above.)
